# Pathways of association between maternal haemoglobin and stillbirth: path-analysis of maternity data from two hospitals in England

Manisha Nair,[1] Marian Knight,[1] Susan Robinson,[2] Catherine Nelson-Piercy,[3] Simon J Stanworth,[4] David Churchill[5]

[1]National Perinatal Epidemiology Unit (NPEU), Nuffield Department of Population Health, University of Oxford, Oxford, UK
[2]Guy's and St Thomas' NHS Foundation Trust, Guy's Hospital, London, UK
[3]Guy's and St Thomas' NHS Foundation Trust, St Thomas' Hospital, London, UK
[4]Oxford University Hospitals NHS Foundation Trust/ NHS Blood and Transplant, John Radcliffe Hospital, Oxford, UK
[5]The Royal Wolverhampton Hospital Hospital NHS Trust, New Cross Hospital, Wolverhampton, UK

**Correspondence to**
Dr Manisha Nair;
manisha.nair@npeu.ox.ac.uk

## ABSTRACT

**Objective** To investigate the mechanisms that link maternal haemoglobin concentration with stillbirth.

**Design** A retrospective cohort analysis using anonymised maternity data from two hospitals in England.

**Setting** The Royal Wolverhampton NHS Trust and Guy's and St Thomas' NHS Foundation Trust.

**Study population** 12 636 women with singleton pregnancies ≥24 weeks of gestation giving birth in the two hospitals during 2013–2015.

**Method** A conceptual framework of hypothesised pathways through birth weight-for-gestational age and maternal infection including potential confounders and other risk factors was developed and examined using path-analysis. Path-analysis was performed by fitting a set of regression equations using weighted least squares adjusted for mean and variance. Goodness-of-fit indices were estimated.

**Main outcome measures** Coefficient of association (β) for relationship between each parameter, and direct, indirect and total effects via the postulated pathways.

**Results** The path-model showed a significant adjusted indirect negative effect of maternal haemoglobin on stillbirth mediated via birth weight-for-gestational age (standardised estimate (SE)=−0.01; 95% CI=−0.01 to −0.001; P=0.028). The effect through maternal infection was not significant at P<0.05 (SE=0.001; 95% CI=−0.004 to 0.01; P=0.610). There was a residual direct negative effect of maternal haemoglobin on stillbirth (SE=−0.12; 95% CI −0.23 to −0.02; P=0.020) after accounting for the two pathways. Total indirect SE=−0.004; 95% CI −0.01 to 0.003; P=0.267; total direct and indirect SE=−0.13; 95% CI −0.23 to −0.02; P=0.016. The goodness-of-fit indices showed a good fit between the model and the data.

**Conclusion** While some of the influence on risk of stillbirth acts through low birth weight-for-gestational age, the majority does not. Several new mechanisms have been suggested for how haemoglobin may be exerting its influence on the risk of stillbirth possibly involving genetic, epigenetic and/or alternative obstetric and nutritional pathologies, but much more research is needed.

## INTRODUCTION

Stillbirth is a global problem, and while the rate has been gradually falling it is difficult

### Strengths and limitations of this study

► While a number of studies have demonstrated low maternal haemoglobin to be a risk factor for stillbirth, this study advances the knowledge about the relationship between maternal haemoglobin and stillbirth by delineating the pathways using a statistical modelling technique called path-analysis.
► Path-models are not causal models and therefore the findings of this study are hypothesis-generating rather than demonstrating causal pathways.
► Inability to adjust for socioeconomic status in the main model was a limitation, but sensitivity analysis did not materially change the results.
► Information about the causes of stillbirth in the study population was not available and it is possible that the mechanisms through which haemoglobin affects stillbirth vary by cause.

to discern the impact of different initiatives. In high-income countries, the rates have only marginally improved. Many of the risk factors associated with stillbirth are known and include maternal obesity, advanced maternal age, smoking, small-for-gestational age (SGA) fetuses, placental abruptions, placental pathology, pre-existing diabetes and hypertension.[1] Many initiatives have been and are being deployed to try to modify these risk factors and to reduce the rate of stillbirth further, but so far, the improvements have been small, for a combination of reasons. Risk factors such as smoking and obesity are hard to modify in a short period of time and ideally, to have a significant impact, need to be influenced in the periconception period. Likewise, in the instance of pre-existing diabetes, it has been shown that preconception blood glucose control is as important as good blood glucose control during pregnancy.[2 3] Other risk factors are pathologically non-specific. SGA refers to a subpopulation of fetuses below a set population centile, usually

the tenth. Within this group is a subset of fetuses that are truly growth restricted, due to placental pathology. These fetuses are at increased risk of adverse obstetric outcome, including stillbirth. In order to identify these 'sick' fetuses, tests of fetal well-being need to be performed, for example, Doppler studies of the placental and fetal circulations. While sufficient progress has been made in using ultrasound to identify the genuinely compromised fetus, it is still not 100% sensitive or specific. Risk factors such as fetal growth restriction and abruption are at or near the end point of the natural history of disease and are a manifestation of established placental pathology. As such, modifying the processes that lead to fetal death from these stand points is difficult. The main and often only intervention available to the obstetrician is to deliver the baby, which is an intervention that can cause significant morbidity and rarely mortality, for either the mother and/or fetus. As a result, calls have been made to improve the understanding of the epidemiology and the causal pathways of stillbirth so that interventions can be targeted at points where the causes of stillbirth are amenable to modification.[4]

We recently studied the association between maternal anaemia and stillbirth in a cohort of 14 001 pregnant women.[5] The cohort was drawn from two inner city populations in England. After adjusting for 11 known confounding variables, the risk of stillbirth decreased linearly per unit (10 g/L) increase in haemoglobin concentration measured in the first trimester visit between 9 and 12 weeks (adjusted OR 0.70, 95% CI 0.58 to 0.85). Compared with women who had a haemoglobin

concentration of over 110 g/L in the first trimester, the risk of stillbirth was five fold higher in women with moderate-to-severe anaemia, <100 g/L. The objective of this study was to further investigate the mechanisms that link maternal haemoglobin concentration in the first trimester of pregnancy with stillbirth.

## METHOD
### Study design
We conducted a retrospective cohort analysis using anonymised maternity data from 14 001 women with singleton pregnancies ≥24 weeks of gestation giving birth in two hospitals between 2013 and 2015 (7175 from Royal Wolverhampton NHS Trust, 2013–2014 and 6826 from Guy's and St Thomas' NHS Foundation Trust, 2014–2015). Information on maternal haemoglobin concentration at the first visit, usually between 9 and 12 weeks' gestation was extracted from the hospital laboratory databases and then paired with maternity data. The datasets were then anonymised and analysis was restricted to 12 636 singleton babies born after 24 weeks of gestation and for whom information about haemoglobin at first visit and infant outcomes was available. The outcome 'stillbirth' was defined as 'the death of a baby occurring before or during birth once a pregnancy has reached 24 weeks'.[6] A theoretical conceptual framework of the hypothesised pathways through which maternal haemoglobin could be associated with stillbirth was developed using Directed Acylic Graphs[7 8] (shown in figure 1). This was subsequently tested using a statistical modelling technique known as path-analysis.

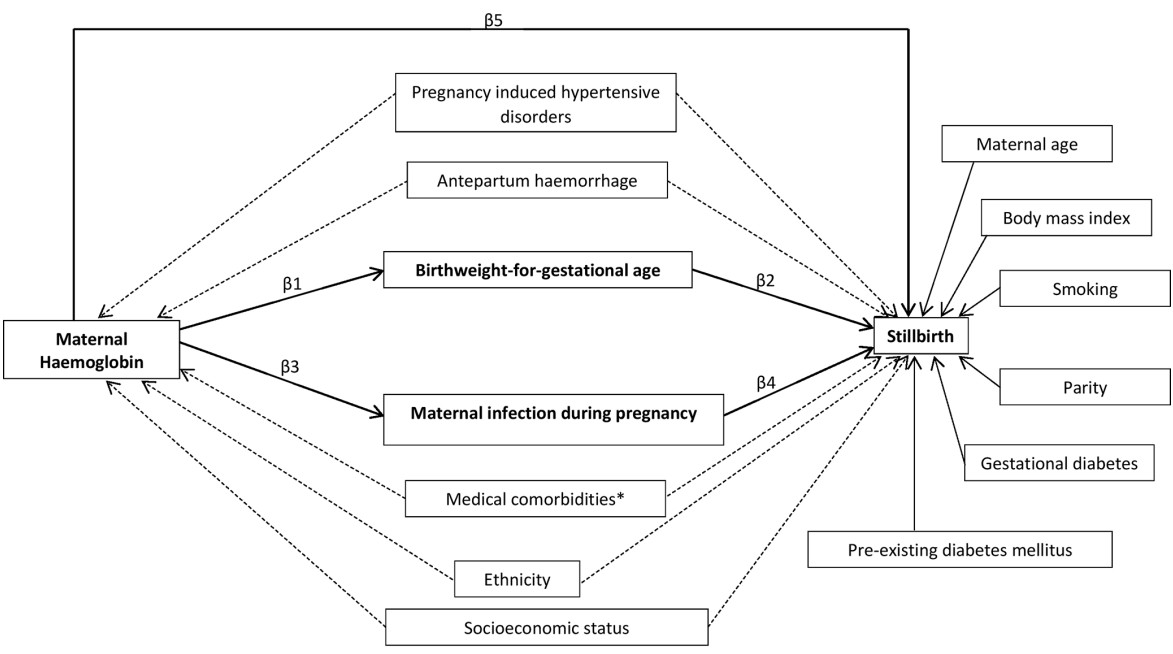

*Medical comorbidities other than pre-existing diabetes mellitus.

The tested pathways are highlighted in bold and confounders are shown using dotted arrows.

β – denotes the 'coefficient of association'

**Figure 1**  Theoretical conceptual framework of hypothesised pathways of effect of maternal haemoglobin on stillbirth.

## Conceptual framework

It has been shown that causal pathways for stillbirth involve fetal growth restriction.[9] Several studies show that maternal haemoglobin concentration is inversely associated with SGA in pregnant women with anaemia.[10 11] Also, low haemoglobin concentration is known to be associated with increased risk of infection during pregnancy and delivery. Therefore, it was hypothesised that the observed effect of haemoglobin concentration on stillbirth could be potentially mediated via birth weight-for-gestational age (an indicator for fetal growth restriction) and/or maternal infection during the index pregnancy. Major factors identified from the literature that could confound the association between maternal haemoglobin and stillbirth were pregnancy-induced hypertensive disorders, ethnicity, antepartum haemorrhage, low socioeconomic status and medical comorbidities. In addition, although not directly associated with maternal haemoglobin, factors such as smoking, high body mass index (BMI), nulliparity, advanced maternal age (>35 years), gestational diabetes and pre-existing diabetes mellitus[1 9 12] are also important as risk factors for stillbirth and therefore were included in the conceptual framework (figure 1).

## Study variables

Sociodemographic characteristics, BMI, obstetric history, current pregnancy problems and medical comorbidities were used to generate the study variables. Based on reported ethnic background, women were divided into 'white' and 'non-white' groups. Information on ethnicity was not available for about 16% of the study sample. Women with unknown ethnicity were included in the 'white group', as has been performed previously[13] because the redistributed proportions matched more accurately with the estimated ethnic profiles in the UK population census. Maternal records relating to problems during the index pregnancy were used to generate binary variables for antepartum haemorrhage, gestational diabetes and hypertensive disorders of pregnancy. Three binary variables were generated from the history of medical comorbidities; pre-existing haemoglobinopathies, pre-existing diabetes mellitus and any other medical comorbidities (excluding obesity).

We calculated the z-scores for birth weight-for-gestational age using the LMS-Growth tool that uses Microsoft Excel add-in written using Excel 2000 with Visual Basic for Applications based on LMS method and the 1990 British reference cohort. This method adjusts for sex and gestational age while calculating the z-scores. The z-scores were converted to centiles using a standard formula in Excel.

## Statistical analysis

We conducted an initial examination of the relationships between the key individual components of the hypothesised pathways (maternal haemoglobin, birth weight-for-gestational age, maternal infection and stillbirth) using unadjusted linear and logistic regression analysis. Test for deviations from linearity using fractional polynomials did not suggest the presence of significant non-linear associations between haemoglobin at first visit and birth weight-for-gestational age centiles or between maternal infection and haemoglobin concentration. We tested the correlation between the other factors included in the conceptual framework. The calculated pairwise correlation coefficients did not show any statistically significant moderate or strong correlations among the factors. We tested for plausible interactions between haemoglobin concentration and mother's ethnicity, haemoglobin and BMI, and birth weight-for-gestational age and ethnicity by fitting interaction terms into each of the univariable models that tested the crude associations between the individual pathway components followed by likelihood ratio testing. No significant interactions were observed.

We conducted path-analysis[14] to examine the pathways of effect of haemoglobin concentration on stillbirth guided by the theoretical conceptual framework (figure 1). Path-analysis was performed by fitting a set of regression equations under the assumption that the model is not affected by unmeasured confounding.[14] Weighted least squares adjusted for mean and variance was used to estimate the parameters for the model.[15] This estimator with pair-wise deletion is considered to be an efficient and unbiased estimator for models with missing data.[16] Missing information was <2% for most variables, except for BMI and smoking. Three goodness-of-fit indices, Comparative Fit Index, $\chi^2$ test for model fit and root mean square error of approximation, each related to a specific aspect of the model were used to quantify the degree of correspondence between the model and the data.[17 18] Indirect effects were computed by multiplying the relevant path coefficients. Statistical significance was considered at the 5% level and the analysis was performed using Mplus V.7.

## Sensitivity analysis

Data on index of multiple deprivation (IMD) quintiles, a measure of socioeconomic status, were available only in the Wolverhampton dataset, hence path-analysis was repeated using these data to measure the effect of IMD quintiles on the hypothesised pathways by testing two models, one with IMD quintiles in addition to the other 11 variables and one without. The results did not vary with the inclusion and exclusion of the variable.

## Patient and public involvement

This is not applicable since the study was a secondary analysis of anonymous hospital data.

## RESULTS

In total, 76 babies were stillborn in the study population. Details of the characteristics of the study population and their comparison with that of the general population of pregnant women in England are described in a previous

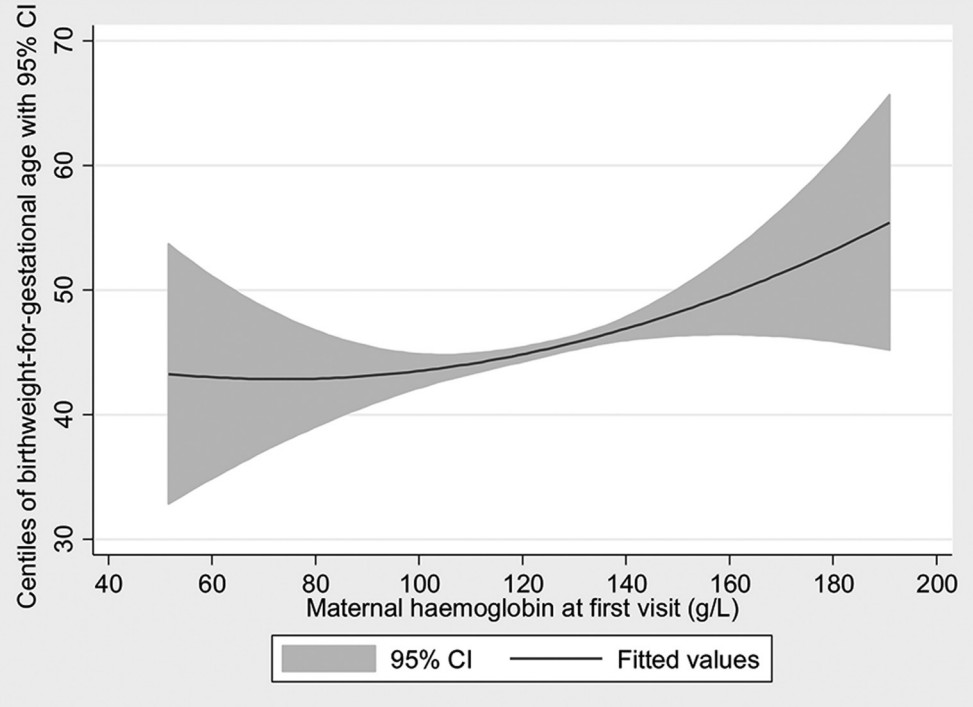

**Figure 2** Association between birth weight-for-gestational age centiles and haemoglobin at first visit.

paper.[5] Briefly, the median age of pregnant women was 30 years (range 14–53 years) and median BMI was 25 kg/m² (range 10–74 kg/m²). Nearly half of the women were multiparous (48%), 13% smoked during pregnancy and 30% belonged to ethnic minority groups. A quarter of the women had one or more pre-existing medical problems, 0.4% had antepartum haemorrhage, 5% were diagnosed with gestational diabetes, 5% had hypertensive disorders of pregnancy and about 7% had other problems during the index pregnancy.

As shown in figure 2, there was a statistically significant crude positive linear association between maternal haemoglobin and centiles of birth weight-for-gestational age (coefficient of association β=0.09; 95% CI 0.04 to 0.13; P<0.001) and the crude odds of stillbirth decreased by 3% per centile increase in birth weight-for-gestational age (OR 0.97; 95% CI 0.96 to 0.98; P<0.001) (figure 3). With regard to the components of the second pathway (association between maternal haemoglobin and stillbirth mediated through maternal infection), the crude odds of maternal infection during current pregnancy decreased linearly per unit increase in haemoglobin concentration (OR 0.99; 95% CI 0.98 to 1.00; P=0.026), but the crude odds of stillbirth did not vary significantly by the presence of maternal infection (OR 0.36; 95% CI 0.05 to 2.58; P=0.309).

The results of the path-analysis are shown in figure 4 and coefficients for the direct and indirect pathways are summarised in table 1. The parameter estimates are the probability coefficients (β), and their magnitude and direction demonstrate the inter-relationships between the variables included in the pathway. As hypothesised, the path-model showed a significant indirect negative

effect of maternal haemoglobin at first visit on stillbirth via birth weight-for-gestational age, although the coefficient of association is small. After controlling for potential confounders, a 1 SD increase in haemoglobin concentration resulted in 0.01 SD decrease in stillbirth mediated via birth weight-for-gestational age. The hypothesised pathway of effect through maternal infection was not significant at P<0.05. After accounting for the effects through the two hypothesised pathways, there was still a significant direct negative effect of maternal haemoglobin on stillbirth (β5=−0.12; 95% CI −0.23 to −0.02; P=0.020). In total (direct and indirect effects), a 1 SD increase in haemoglobin concentration resulted in 0.13 SD decrease in stillbirth (P=0.020). In addition, we observed significant indirect effects of other known risk factors such as parity, BMI, smoking, gestational diabetes and pre-existing diabetes mellitus on stillbirth via their effects on birth weight-for-gestational age. These associations have been shown in other studies and therefore further validates our model. Pregnancy-induced hypertensive disorders and ethnicity were significant confounders. The goodness-of-fit indices showed a good fit between the model and the data.

## DISCUSSION
The causal pathways for stillbirth are complex often with multiple risk factors interacting to influence the eventual outcome. We postulated that two pathways were most likely to mediate the effect of maternal haemoglobin during the first trimester on stillbirth: birth weight-for-gestational age and maternal infection. Of these, only birth weight-for-gestational age was found

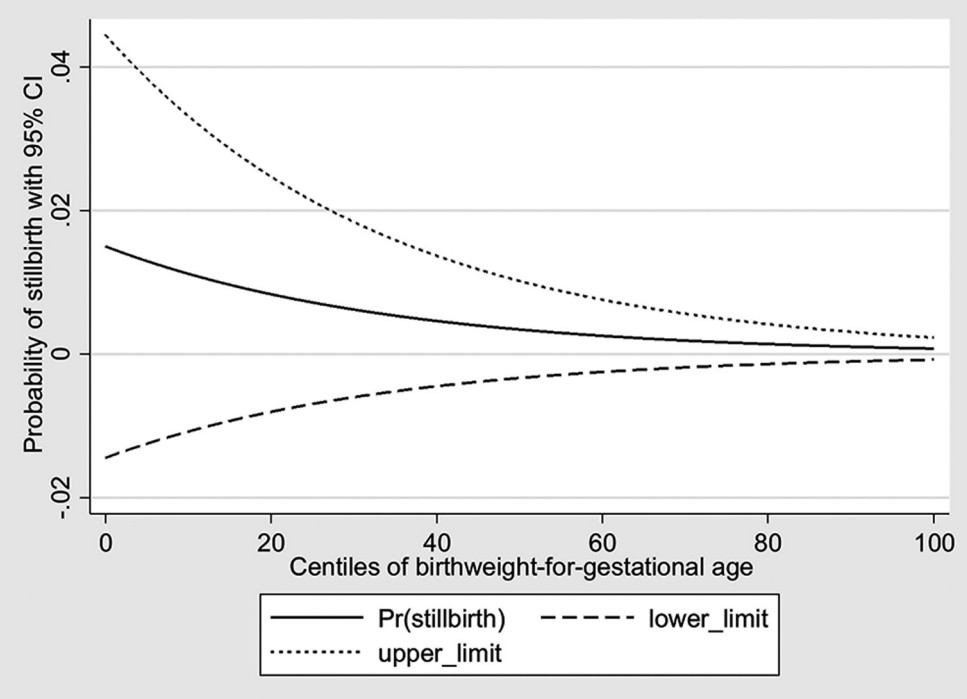

**Figure 3** Association between birth weight-for-gestational age centiles and stillbirth.

to be statistically significantly mediating the effect. Neither pathway completely explained the effect of haemoglobin on stillbirth as there was a significant residual direct effect of haemoglobin in the first trimester on stillbirth. This suggests that there are other unidentified factors involved in the pathway(s).

While a number of studies have demonstrated low maternal haemoglobin or maternal anaemia to be a risk factor for stillbirth,[5 10 19–21] this study went a step further in delineating the pathways through which maternal haemoglobin could affect stillbirth. In addition to testing known pathways, our study showed that several mechanisms are still unknown and need further investigation. However, it is important to acknowledge that path-models are not causal models and therefore the findings of our study are hypothesis-generating rather than confirmed causal pathways. Inability to adjust for socioeconomic status in the main model was a limitation, but sensitivity analysis using the Wolverhampton data did not materially change the results. We did not have information about the causes of stillbirth in the study population and it is possible that the mechanisms through which haemoglobin affects stillbirth vary by cause. For example, pathways of effect for stillbirth due to congenital anomalies could be different from the pathways for stillbirth as a result of fetal growth restriction.

The observed pathway of effect of maternal haemoglobin on stillbirth mediated via low birth weight-for-gestational age could be explained by a number of factors. In addition to the possibility of haemoglobin exerting its influence through a reduced oxygen tension, there could be other plausible mechanisms through its interaction with nitric oxide (NO), carbon monoxide (CO)

and carbon dioxide ($CO_2$) affecting placental circulation leading to fetal growth restriction resulting in stillbirth in some cases.[22 23] However, these factors need to be explored further to generate evidence of biologically plausible mechanisms. It is known that anaemia per se increases the risk of infection, but iron supplementation in iron replete women has also been associated with infectious causes of stillbirth.[24] In our study, the pathway to stillbirth via maternal infection showed no significant relationship leading us to conclude that the haemoglobin effect on stillbirth was not mediated through infection. However, our data included only overt infections reported during pregnancy and it is possible that subclinical infections could influence the pathway.

After accounting for the two hypothesised pathways, known confounders and risk factors for stillbirth, there was still a residual direct effect of haemoglobin on stillbirth. This suggests that the relationship between maternal haemoglobin at first trimester and stillbirth cannot be explained with what we already know. This presents the prospect of new and novel mechanisms through which haemoglobin may be affecting the risk of stillbirth. The formation of the placenta from the earliest stages of pregnancy is a highly dynamic process. The early conceptus is a largely hypoxic environment and it is possible that its transition to an oxygen-rich environment with a fully functional placenta is adversely affected by low haemoglobin, either through deficient oxygen delivery to the relevant tissues or through maladaptation of the NO-driven vascular redistribution through vasodilatation in the presence of hypoxia. However, as well as affecting vascular tone NO has two other important functions: 1) influencing cell signalling and cellular interactions and

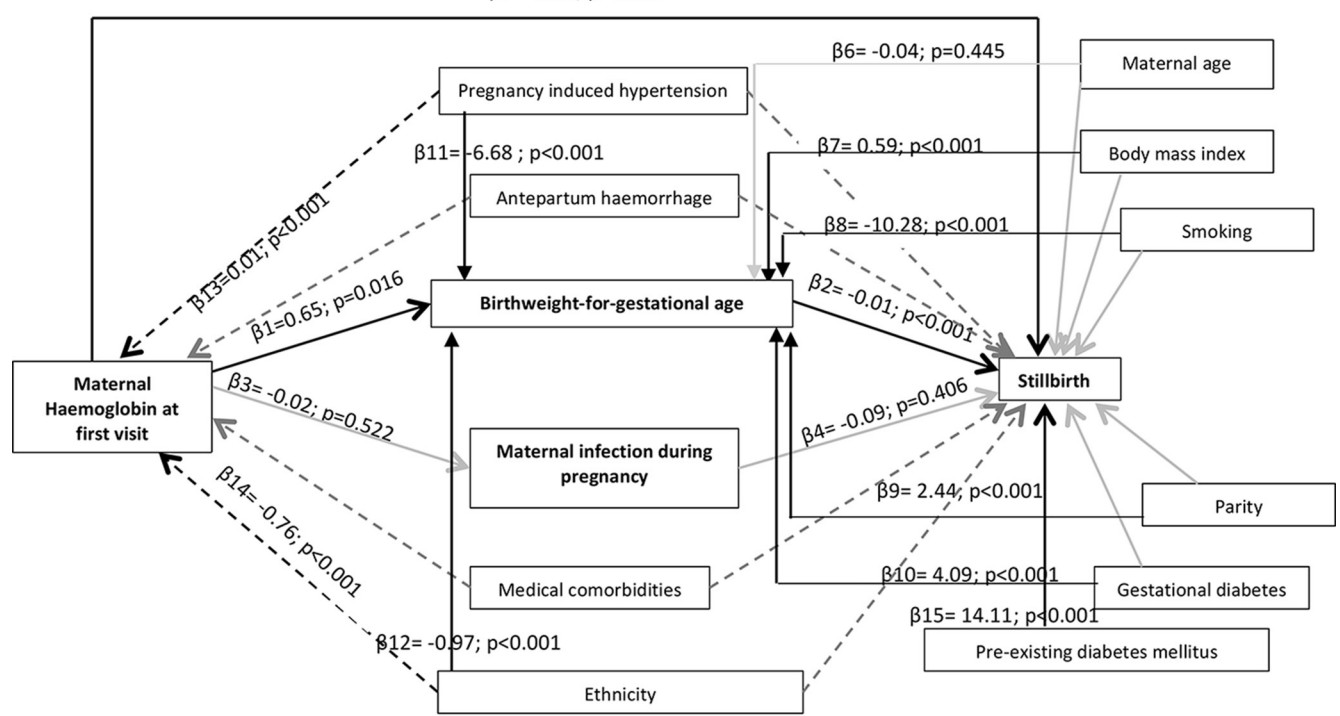

Indirect effect of haemoglobin concentration at first visit on stillbirth via birthweight-for-gestational age: Standardised estimate = -0.01; 95% CI = -0.01 to -0.001; p=0.028. Indirect effect of haemoglobin concentration at first visit on stillbirth via maternal infection: Standardised estimate= 0.001; 95% CI = -0.004 to 0.01; p=0.610. Total indirect effect of haemoglobin concentration at first visit on stillbirth: Standardised estimate= -0.004; 95% CI = -0.01 to 0.003; p=0.267

Total direct and indirect effect of haemoglobin concentration at first visit on stillbirth: Standardised estimate = -0.13; 95% CI = -0.23 to -0.02; p=0.016

P-value for χ2 test for model fit <0.001; RMSEA = 0.00, 90% CI 0.00 to 0.02; CFI = 1.00; R-Square for stillbirth = 0.09

**Figure 4** Path-model showing the association between maternal haemoglobin and stillbirth. CFI, Comparative Fit Index; RMSEA, root mean square error of approximation.

2) neural function.[25] The first of the two could lead to pathologies that are currently classed histologically as villous dysmaturity, sometimes seen with some normally grown stillbirths, rather than the vasculopathy associated

with pre-eclampsia and growth-restricted fetuses. Further research matching placental histological findings to cellular function may help to reveal molecular and functional abnormalities that are also critical to normal placental development and fetal survival.

An altogether more prosaic explanation is that the mother's concentration of haemoglobin, or more specifically low haemoglobin, could be a marker for another 'abnormality' (eg, undiagnosed inflammatory conditions, autoimmune disease, renal disease, nutritional deficiencies, etc). Iron itself plays a pivotal role in several metabolic processes and a deficiency at any time during a pregnancy may confer a disadvantage on the woman and/or the fetus. Low iron stores, leading to iron-deficient anaemia, may be an indicator of poor nutrition and deficiencies in other micronutrients, which either alone or in concert may play a role in the increased risk

**Table 1** Direct and indirect pathways of association between maternal haemoglobin at first trimester and stillbirth

| Pathways | Coefficient (SE) | P value |
|---|---|---|
| Direct | −0.125 (0.054) | 0.020 |
| Total indirect | −0.004 (0.004) | 0.267 |
| Total direct and indirect | −0.129 (0.054) | 0.016 |
| **Specific indirect pathways** | | |
| Via birth weight-for-gestational age | −0.005 (0.002) | 0.028 |
| Via maternal infection | 0.001 (0.003) | 0.610 |

of stillbirth.[26][27] Finally, epigenetic phenomena affecting gene expression in the maternal genome could be exerting a significant influence on placental and fetal development in the first trimester. The imprinting or silencing of some genes through epigenetic mechanisms may adversely affect the foundations laid down in the first trimester and lead to an as yet unrecognised higher risk pregnancy. Iron is a major cofactor for many metabolic processes including imprinting through methylation of sequences of DNA. Other epigenetic mechanisms such as templating (structural changes to cell membranes), or interfering with RNA silencing also have critical roles to play. Alternatively, rather than iron, these epigenetic mechanisms may be affected by haemoglobin itself via its other suggested functions such as a NO donor.[28]

While the antecedents of stillbirth are well-known, the mechanisms through which they exert their effect on this outcome still remain unclear. Our findings clearly show that while some of the influence from haemoglobin concentration on risk of stillbirth acts through low birth weight-for-gestational age possibly as an adjunct to vascular pathology in the placenta, the majority does not. Haemoglobin may be exerting its influence on the risk of stillbirth involving genetic, epigenetic and/or alternative obstetric and nutritional pathologies, but more research needs to be undertaken to understand these relationships. Our findings suggest that prevention of anaemia will also have a beneficial impact on birth weight which in turn could influence favourably the intergenerational risk of stillbirth. However, more research needs to be performed on causal mechanisms, if we are to understand in depth the pathologies through which maternal haemoglobin affects pregnancies and fetal outcomes.

**Acknowledgements** The authors would like to acknowledge the contribution of the following people from the Royal Wolverhampton NHS Trust: Alain Rolli, Clinical Scientist, for extracting the haematological data; Laura Gardiner, Clinical Trials Coordinator, Katherine Cheshire, Research Midwife and Julia Icke, Research Midwife, for validating the clinical and haematological data; Bernie Williams IT midwife for extracting the obstetric clinical data. The authors would also like to thank Marcelo Canda, Business Information Analyst, Women's Services, Guy's and St. Thomas' NHS Foundation Trust for helping with extracting and merging the clinical and haematological data.

**Contributors** MN designed the study, carried out the data analysis, interpreted the data and wrote the first draft of the manuscript. DC designed the study, facilitated the process of data extraction from the hospital records, contributed to the data analysis plan and interpretation of the results and edited the manuscript. SR facilitated the process of data extraction from the hospital records, contributed to interpretation of the results and edited the manuscript. CN-P contributed to interpretation of the results and edited the manuscript. SJS designed the study, contributed to the data analysis plan and data interpretation and edited the manuscript. MK designed the study, contributed to the data analysis plan, data interpretation and edited the manuscript.

**Funding** MK is funded by a National Institute for Health Research (NIHR) Research Professorship (NIHR-RP-011-032).

**Disclaimer** The views expressed in this publication are those of the author(s) and not necessarily those of the NHS, the NIHR or the Department of Health. The funding sources had no role in the study, and the researchers were independent from the funders. The funders had no role in study design, data collection and analysis, decision to publish or preparation of the manuscript.

**Competing interests** None declared.

**Patient consent** Not required.

**Provenance and peer review** Not commissioned; externally peer reviewed.

**Data sharing statement** There are no unpublished data from this study. To access the data, please contact the authors: SR and DC.

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
