## [Reviewer comments · BMJ Open]

ARTICLE DETAILS

TITLE (PROVISIONAL)	Pathways of association between maternal haemoglobin and stillbirth: path-analysis of maternity data from two hospitals in England
AUTHORS	Nair, Manisha; Knight, Marian; Robinson, Susan; Nelson-Piercy, Catherine; Stanworth, Simon; Churchill, David

VERSION 1 – REVIEW

REVIEWER	Svein Rasmussen Institute of Clinical Medicine, University of Bergen, Noray
REVIEW RETURNED	05-Nov-2017

GENERAL COMMENTS	The purpose of the study, "... to further investigate the mechanisms that link maternal haemoglobin concentration in the first trimester of pregnancy with stillbirth", as stated in the end of the Introduction section is clear enough. In lines 23-28, page 4, the authors state that "Small for gestational age, as applied to the fetus, selects a population who are considered at risk. It does not detect the 'sick' fetus. Other tests such as assessments of fetal wellbeing need to be employed to do that, but they too have their diagnostic limitations." What actually is assessed by serial measures in obstetrical practice is fetal growth, most often caused by placental insufficiency, which may be caused or aggravated by chronic disease. Small for gestational age is only a rough proxy of fetal growth restriction and may include both constitutional small fetuses and growth restricted fetuses. The statement that "Other tests such as assessments of fetal wellbeing need to be employed to do that, but they too have their diagnostic limitations" is not necessarily correct because during the last decades there has been considerable progress in fetal surveillance, such as Doppler techniques. The outcome, stillbirth is not clearly defined. In the 1st paragraph of the Method section and Figure 1 the authors outline the conceptual framework of 1st trimester haemoglobin and stillbirth. In the manuscript, small for gestational age should be substituted with fetal growth restriction where appropriate. Per reading, it is obscure what the βs in the figure mean. Lines 40-41, 1st paragraph: "...although not directly associated with maternal haemoglobin, factors such as smoking, "Smoking does indeed have effect on maternal haemoglobin through formation of methaemoglobin. Other mechanisms are also involved in the association between smoking and fetal growth restriction or fetal death. Additionally, smoking increases the distance between
--

	maternal and fetal blood, thus affecting the effect of maternal haemoglobin on fetal nutrition. Consistently, It is known from the literature that smoking is associated with maternal haemoglobin throughout pregnancy. The association found in the literature between 1st trimester haemoglobin and fetal growth restriction does thus not agree with the framework in the figure. Lines 56-57 page 5: "Women with unknown ethnicity were included in the 'white group',...". This is not a big issue, but why are not women with unknown ethnicity included in an own category? In the Statistical analysis subsection the authors state "We did not find any significant moderate to strong correlations among the other factors included in the conceptual framework". This is should be made clearer. Otherwise the subsection is adequate. The Results section lacks tables which could have improved the readability of the manuscript. The characteristics in the 1st paragraph could be placed in a table. In the 2nd paragraph of Results the authors state that "... there was a statistically significant crude positive linear association between maternal haemoglobin and birthweight-for-gestational age" (Figure 2). Actually, the Figure 2 shows the association between 1st trimester haemoglobin and birthweight centiles for gestational age. How were the centiles calculated? Why were not standard deviation scores used in Figures 2 and 3? In Figure 3 centiles are in the exposure. Taken together, Figures 2 and 3 shows the association between maternal haemoglobin and stillbirth with birthweight centiles (a proxy of fetal growth) as an intermediate variable. Same paragraph, lines 35-41 "With regard to the components of the second pathway, the crude odds of maternal infection during index pregnancy decreased linearly per unit increase in haemoglobin concentration (OR=0.99; 95% CI 0.98 to 1.00; p=0.026), but the crude odds of stillbirth did not vary significantly by the presence of maternal infection (OR= 0.36; 95% CI 0.05 to 2.58; p=0.309)." The sentence is unclear. The meanings of the "second" pathway and "the index pregnancy" are not obvious. In the Discussion section, the authors acknowledge the important limitation of the study that information on causes of fetal death was lacking. The only statistically significant pathway was 1st trimester maternal haemoglobin via birthweight for gestational age. Thus how the variable birthweight for gestational age was calculated is important, e.g. whether it was parity and gender specific.
--	---

REVIEWER	Alan N Schechter National Institutes of Health Bethesda, Maryland 20892 U.S.A.
REVIEW RETURNED	17-Nov-2017

GENERAL COMMENTS	This paper uses a sophisticated statistical methodology (pathway analysis) to attempt to delineate better the factors that contribute to stillbirth (defined as death after >24 weeks of gestation) for more than 10,000 women attending maternity clinics in two UK hospitals in the 2013 to 2015 period. In particular this group has just now published in the BMJ of an association of maternal anemia with marked increases in stillbirth frequency in this group of patients (with
--

	76 stillbirths in the two years) and now test further the relationship of hemoglobin level at first visit to stillbirth, as such a relationship has been reported before by others as well, but with some uncertainties. The new methods confirm the previous finding and show that the effects of hemoglobin levels on birthweight for gestational age and observed maternal infections could not account for all the effects of variations of the initial hemoglobin levels, although socio-economic status could not be fully quantitated. The authors discuss possible ways in which maternal hemoglobin might affect the fetus although they do not present any mechanistic data to test such possibilities and note the need for much further research on this public health issue. The hospital work is done in collaboration with the excellent NPEU group at Oxford for the statistical analyses and I would trust their approaches, although I would hope that this journal would obtain relevant expertise about these methods, in reviewing this paper, about which I have limited knowledge. On the other hand I think this application of "big data" is an important advance in medicine and may lead to new concepts related to this important clinical problem as well as to the mechanisms involved.
--	---

REVIEWER	Dr Victoria Allgar University of York, England
REVIEW RETURNED	08-Jan-2018

GENERAL COMMENTS	Data had already been published on the association between maternal anaemia and stillbirth (ref 5). The objective of this study was to further investigate the mechanisms that link maternal haemoglobin concentration in the first trimester of pregnancy with stillbirth using path-analysis. In the statistical analysis section it states that "We conducted an initial examination of the relationships between the key individual components of the hypothesised pathways." The results of this analysis is not presented in the paper - Could this be included as a supplementary table? The paper reports the path-analysis and this seems appropriate with correct interpretation. Path analysis is a straightforward extension of multiple regression. Its aim is to provide estimates of the magnitude and significance of hypothesised causal connections between sets of variables.
---

VERSION 1 – AUTHOR RESPONSE

Comments from the Associate Editor:

Just reading the cover letter was enough to convince me that this adds something, but they could do a better job putting this in the context of previously published research ,in particular their recent paper, which is listed as being "in press" (reference 5), but which seems to have been published already: rBr J Haematol. 2017 Dec;179(5):829-837. doi: 10.1111/bjh.14961. Epub 2017 Oct 26. Association between maternal haemoglobin and stillbirth: a cohort study among a multi-ethnic population in England. <https://www.ncbi.nlm.nih.gov/pubmed/29076149>

They really only say in the discussion section "While a number of studies have demonstrated low maternal haemoglobin or maternal anaemia to be a risk factor for stillbirth", but then don't even cite any studies. They also don't cite some studies I easily picked up:

Acta Obstet Gynecol Scand. 2016 May;95(5):555-64. doi: 10.1111/aogs.12862. Epub 2016 Mar 1.
Maternal and neonatal outcomes of antenatal anemia in a Scottish population: a retrospective cohort study. <https://www.ncbi.nlm.nih.gov/pubmed/26846870>
Paediatr Perinat Epidemiol. 2014 Sep;28(5):372-80. doi: 10.1111/ppe.12134. Epub 2014 Jun 17.
Adverse perinatal outcomes associated with moderate or severe maternal anaemia based on parity in Finland during 2006-10. <https://www.ncbi.nlm.nih.gov/pubmed/24938307>

Response: We thank the Editor for suggesting these references. We have added references to the sentence in the discussion section. We have also updated the reference to our BJH paper which has been published.

Editorial Requirements:

- Please revise the Strengths and Limitations section (after the abstract) to focus on the methodological strengths and limitations of your study rather than summarizing the results.

Response: We have revised the 'Strengths and Limitations' section, as suggested.

Reviewer(s)' Comments to Author:

Reviewer: 1

Reviewer Name: Svein Rasmussen

Institution and Country: Institute of Clinical Medicine, University of Bergen, Norway Please state any competing interests: Obstetrics, Perinatal epidemiology

Please leave your comments for the authors below

The purpose of the study, "... to further investigate the mechanisms that link maternal haemoglobin concentration in the first trimester of pregnancy with stillbirth", as stated in the end of the Introduction section is clear enough.

In lines 23-28, page 4, the authors state that "Small for gestational age, as applied to the fetus, selects a population who are considered at risk. It does not detect the 'sick' fetus. Other tests such as assessments of fetal wellbeing need to be employed to do that, but they too have their diagnostic limitations." What actually is assessed by serial measures in obstetrical practice is fetal growth, most often caused by placental insufficiency, which may be caused or aggravated by chronic disease. Small for gestational age is only a rough proxy of fetal growth restriction and may include both constitutional small fetuses and growth restricted fetuses. The statement that "Other tests such as assessments of fetal wellbeing need to be employed to do that, but they too have their diagnostic limitations" is not necessarily correct because during the last decades there has been considerable progress in fetal surveillance, such as Doppler techniques.

Response: We thank the reviewer for this comment and have altered the text accordingly. On page 4 we have edited the text between lines 13 and 22 to address the comment. We have also made minor amendments to the text on pages: 5 (conceptual framework), 10 and 11.

We have continued with birthweight for gestational age in the other areas of the text, because our analysis used this as a continuous variable. Fetal growth restriction being a diagnosis would have to be treated as a categorical variable. It also affects only a proportion of stillbirths that result from specific placental pathologies. We feel that despite its limitations, birth weight is affected by a multitude of biological factors from the genetic to the environmental and therefore, when used as an outcome in the context of our paper, points to the multifarious influences / causes of stillbirth, rather than focusing on one specific pathological type. We contend that if we are to make further progress in preventing this tragic outcome, it is time to look beyond current diagnostic boundaries and stimulate research in new emerging areas of interest.

The outcome, stillbirth is not clearly defined.

Response: As suggested, we have included a definition for stillbirth in the methods section.

In the 1st paragraph of the Method section and Figure 1 the authors outline the conceptual framework of 1st trimester haemoglobin and stillbirth. In the manuscript, small for gestational age should be substituted with fetal growth restriction where appropriate. Per reading, it is obscure what the β s in the figure mean.

Response: The text has been altered in this section. Also please see the comment above.

As suggested by the reviewer, we have explained β in Figure-1. β – denotes the 'coefficient of association'.

Lines 40-41, 1st paragraph: "...although not directly associated with maternal haemoglobin, factors such as smoking, ...: Smoking does indeed have an effect on maternal haemoglobin through formation of methaemoglobin. Other mechanisms are also involved in the association between smoking and fetal growth restriction or fetal death. Additionally, smoking increases the distance between maternal and fetal blood, thus affecting the effect of maternal haemoglobin on fetal nutrition. Consistently, it is known from the literature that smoking is associated with maternal haemoglobin throughout pregnancy. The association found in the literature between 1st trimester haemoglobin and fetal growth restriction does thus not agree with the framework in the figure.

Response: We thank the reviewer for the comments. We think that the reviewer here is referring to carboxyhaemoglobin (due to carbon monoxide) and not methaemoglobin. Methaemoglobin is either congenital or acquired, relates to iron in the oxidised state, and in normal people the level of methaemoglobin in blood is <1% and as far as we know, it is not affected by smoking. We do not think that smoking should be in the pathway since haemoglobin doesn't influence whether someone smokes or not so cannot mediate the effect of Hb on stillbirth. Smoking tends to increase the haemoglobin concentration not decrease it (Reference: 'The effect of cigarette smoking on haemoglobin levels and anemia screening. JAMA 28 Sept 1990 Vol264 No12 p1556-1559'), so if there was an effect it would mitigate against the finding of an inverse relationship with stillbirth risk. The effect of smoking on fetal growth is thought to be mediated through vascular pathology in the placenta which is a 2nd and 3rd trimester issue as the placenta hasn't formed in the first trimester. Therefore the only thing smoking could be is a confounder, which has been addressed in the analysis.

Lines 56-57 page 5: "Women with unknown ethnicity were included in the 'white group',...". This is not a big issue, but why are not women with unknown ethnicity included in an own category?

Response: We thank the reviewer for the comment. The reason for including women with 'unknown ethnicity' in the 'white group' was based on a justification published in a previous study. Knight et al (2009) included women with unknown ethnicity in the 'white European' group because the re-distributed proportions matched more accurately with the estimated ethnic profiles in the UK population census (NHS maternity statistics).

2006. NHS maternity statistics England 2005-6. Statistical bulletin 2006/08, Leeds, Information Centre for Health and Social Care.

KNIGHT, M., KURINCZUK, J. J., SPARK, P. & BROCKLEHURST, P. 2009. Inequalities in maternal health: national cohort study of ethnic variation in severe maternal morbidities. *BMJ: British Medical Journal*, 338, b542.

We also re-ran the model including 'unknown ethnicity' as a separate category, but this did not change the model outputs.

In the Statistical analysis subsection the authors state "We did not find any significant moderate to strong correlations among the other factors included in the conceptual framework". This should be made clearer. Otherwise the subsection is adequate.

Response: As advised by the reviewer, we have clarified this statement in the revised draft.

The Results section lacks tables which could have improved the readability of the manuscript. The characteristics in the 1st paragraph could be placed in a table.

Response: We thank the reviewer for the comment. As mentioned in the submitted paper, the characteristics of the study population has already been described in detail in the published BJH paper (Table-1 in Nair M, Churchill D, Robinson S, et al. Association between maternal haemoglobin and stillbirth: a cohort study among a multi-ethnic population in England. Br J Haematol 2017;179(5):829-37). We do not think that it will be appropriate to include the same table in this paper. However, we have included a table that summarises the results of the direct and indirect pathways of association in the revised draft.

In the 2nd paragraph of Results the authors state that “.. there was a statistically significant crude positive linear association between maternal haemoglobin and birthweight-for-gestational age” (Figure 2). Actually, the Figure 2 shows the association between 1st trimester haemoglobin and birthweight centiles for gestational age. How were the centiles calculated? Why were not standard deviation scores used in Figures 2 and 3? In Figure 3 centiles are in the exposure. Taken together, Figures 2 and 3 shows the association between maternal haemoglobin and stillbirth with birthweight centiles (a proxy of fetal growth) as an intermediate variable.

Response: We thank the reviewer for the comments. As suggested, we have now reworded the sentence as “.. there was a statistically significant crude positive linear association between maternal haemoglobin and centiles of birthweight-for-gestational age.”

The centiles were calculated from z-scores using an Excel formula. The z-scores for birthweight-for-gestational age was calculated using the LMS-Growth tool that uses Microsoft Excel add-in written using Excel 2000 with Visual Basic for Applications (VBA) based on LMS method and the 1990 British reference cohort. This method adjusts for sex and gestational age while calculating the z-scores. We used centiles instead of the z-scores as this is easier to interpret and more meaningful to the readers.

Same paragraph, lines 35-41 “With regard to the components of the second pathway, the crude odds of maternal infection during index pregnancy decreased linearly per unit increase in haemoglobin concentration (OR=0.99; 95% CI 0.98 to 1.00; p=0.026), but the crude odds of stillbirth did not vary significantly by the presence of maternal infection (OR= 0.36; 95% CI 0.05 to 2.58; p=0.309).” The sentence is unclear. The meanings of the “second” pathway and “the index pregnancy” are not obvious.

Response: We thank the reviewer for the comments. We have explained the meaning of ‘second pathway’ in the revised draft. We have changed ‘index pregnancy’ to ‘current pregnancy’ to make the sentence clearer.

In the Discussion section, the authors acknowledge the important limitation of the study that information on causes of fetal death was lacking.

The only statistically significant pathway was 1st trimester maternal haemoglobin via birthweight for gestational age. Thus how the variable birthweight for gestational age was calculated is important, e.g. whether it was parity and gender specific.

Response: As suggested by the reviewer, we have added a paragraph in the methods section describing how the variable birthweight-for-gestational age was generated.

Reviewer: 2

Reviewer Name: Alan N Schechter

Institution and Country: National Institutes of Health, Bethesda, Maryland 20892, U.S.A.

Please state any competing interests: None declared

Please leave your comments for the authors below

This paper uses a sophisticated statistical methodology (pathway analysis) to attempt to delineate better the factors that contribute to stillbirth (defined as death after >24 weeks of gestation) for more than 10,000 women attending maternity clinics in two UK hospitals in the 2013 to 2015 period. In particular this group has just now published in the BMJ of an association of maternal anemia with

marked increases in stillbirth frequency in this group of patients (with 76 stillbirths in the two years) and now test further the relationship of hemoglobin level at first visit to stillbirth, as such a relationship has been reported before by others as well, but with some uncertainties. The new methods confirm the previous finding and show that the effects of hemoglobin levels on birthweight for gestational age and observed maternal infections could not account for all the effects of variations of the initial hemoglobin levels, although socio-economic status could not be fully quantitated. The authors discuss possible ways in which maternal hemoglobin might affect the fetus although they do not present any mechanistic data to test such possibilities and note the need for much further research on this public health issue.

The hospital work is done in collaboration with the excellent NPEU group at Oxford for the statistical analyses and I would trust their approaches, although I would hope that this journal would obtain relevant expertise about these methods, in reviewing this paper, about which I have limited knowledge. On the other hand I think this application of "big data" is an important advance in medicine and may lead to new concepts related to this important clinical problem as well as to the mechanisms involved.

Response: We thank the reviewer for the comments.

Reviewer: 3

Reviewer Name: Dr Victoria Allgar

Institution and Country: University of York, England Please state any competing interests: None declared

Please leave your comments for the authors below Data had already been published on the association between maternal anaemia and stillbirth (ref 5). The objective of this study was to further investigate the mechanisms that link maternal haemoglobin concentration in the first trimester of pregnancy with stillbirth using path-analysis.

In the statistical analysis section it states that "We conducted an initial examination of the relationships between the key individual components of the hypothesised pathways." The results of this analysis is not presented in the paper - Could this be included as a supplementary table?

Response: We thank the reviewer for the comments. We have presented the results in page 7 and in Figures 2 and 3. We are happy to present the same in a Supplementary table as well, but this will be a repetition. Instead we have added a table that summarises the results of the direct and indirect pathways of association in the revised draft.

The paper reports the path-analysis and this seems appropriate with correct interpretation. Path analysis is a straightforward extension of multiple regression. Its aim is to provide estimates of the magnitude and significance of hypothesised causal connections between sets of variables.

VERSION 2 – REVIEW

REVIEWER	Alan N. Schechter, MD National Institutes of Health Bethesda, MD 20892 United States
REVIEW RETURNED	22-Feb-2018
GENERAL COMMENTS	Revision meets concerns raised in the refereeing process.
REVIEWER	Svein Rasmussen Department of Clinical Science, University of Bergen, Bergen, Norway
REVIEW RETURNED	26-Feb-2018
GENERAL COMMENTS	The authors have adequately addressed my comments and the manuscript may be published as it is.